# Novel SiGe/Si Heterojunction Double-Gate Tunneling FETs with a Heterogate Dielectric for High Performance

**DOI:** 10.3390/mi14040784

**Published:** 2023-03-31

**Authors:** Qing Chen, Rong Sun, Ruixia Miao, Hanxiao Liu, Lulu Yang, Zengwei Qi, Wei He, Jianwei Li

**Affiliations:** 1School of Electronic Engineering, Xi’an University of Posts & Telecommunications, Xi’an 710121, China; 2Nova Product Line, Xi’an Huawei Technologies Co., Xi’an 710065, China

**Keywords:** band-to-band tunneling (BTBT), heterojunction, pocket, heterogate dielectric, DGTFET

## Abstract

In this paper, a new SiGe/Si heterojunction double-gate heterogate dielectric tunneling field-effect transistor with an auxiliary tunneling barrier layer (HJ-HD-P-DGTFET) is proposed and investigated using TCAD tools. SiGe material has a smaller band gap than Si, so a heterojunction with SiGe(source)/Si(channel) can result in a smaller tunneling distance, which is very helpful in boosting the tunneling rate. The gate dielectric near the drain region consists of low-k SiO_2_ to weaken the gate control of the channel-drain tunneling junction and reduce the ambipolar current (I_amb_). In contrast, the gate dielectric near the source region consists of high-k HfO_2_ to increase the on-state current (I_on_) through the method of gate control. To further increase I_on_, an n^+^-doped auxiliary tunneling barrier layer (pocket)is used to reduce the tunneling distance. Therefore, the proposed HJ-HD-P-DGTFET can obtain a higher on-state current and suppressed ambipolar effect. The simulation results show that a large I_on_ of 7.79 × 10^−5^ A/μm, a suppressed I_off_ of 8.16 × 10^−18^ A/μm, minimum subthreshold swing (SS_min_) of 19 mV/dec, a cutoff frequency (f_T_) of 19.95 GHz, and gain bandwidth product (GBW) of 2.07 GHz can be achieved. The data indicate that HJ-HD-P-DGTFET is a promising device for low-power-consumption radio frequency applications.

## 1. Introduction

With continuous improvements in chip integration and application requirements, the dimensions of conventional MOSFETs are continuously scaling down to a few nanometers. This results in a number of critical issues, such as leakage current, high-power dissipation, short channel effects (SCEs), and a sub-threshold slope restriction of 60 mV/decade for conventional MOSFETs at room temperature [1,2,3,4,5]. Moreover, the contradiction between static power consumption and device performance caused by off-state leakage current has become a serious problem.

To overcome the limitations of conventional MOSFETs, many novel device structures have been proposed in the past couple of years. Tunnel field-effect transistors (TFETs), one of the most promising candidates for MOSFETs, have been attracting more and more attention in recent years. Because the high-energy tail states of the Fermi distribution function in the source are removed, the off-state current (I_off_) of TFETs is much lower than that of the MOSFETs [6]. In addition, because band-to-band tunneling (BTBT) is the main operation mechanism in TFETs, they can break the limitation of 60 mV/decade subthreshold swing (SS) in conventional MOSFETs, which relies on hot electron emission, especially at low voltages [7,8]. Finally, TFETs are more immune to short channel effects and temperature variations [9,10]. Therefore, these advantages make TFETs more favorable for circuits that are based on low-power applications.

However, the main problems of TFETs are low on-state current (I_on_), high off-state current (I_off_), and ambipolar behavior, which are the main challenges that traditional TFETs face [11,12,13]. The driven current of the conventional TFETs is determined by the gate-modulated tunneling diode. Furthermore, for the silicon-based TFET, the large bandgap and carrier mass in silicon material result in a small drive current. Thus, double-gate tunnel field-effect transistors (DGTFETs) are expected to extend the limitations of leakage current and subthreshold slope [10,14,15,16]. At present, various methods, such as gate dielectric engineering [17,18,19,20], hetero-structure [21,22,23,24], dielectric pocket [25,26,27,28,29] and work function engineering [15,30,31,32], are proposed to improve the I_on_. However, most of these approaches significantly boost I_on_ at the expense of other device performance metrics, such as I_off_, ambipolar current, and analog/RF performance.

Going forward, silicon germanium (SiGe) will most likely be adopted for future low-power very-large-scale integration (VLSI) technologies among all material systems for TFET applications, owing to its VLSI compatibility, mature synthesis techniques, and tunable bandgap [11,33,34]. In this work, we propose a new Si_1-x_Ge_x_/Si heterostructure TFET with high-k/low-k dual material heterogeneous gate dielectric and a heavily doped pocket within the channel (HJ-HD-P-DGTFET). The use of p-type doped SiGe material in the source region instead of adopting germanium and III-V compound semiconductors in the source region of TFET can lead to higher ON/OFF ratios due to a smaller tunneling distance obtained by the heterojunction, which is very helpful to boost the tunneling rate [15]. Furthermore, an n^+^-doped auxiliary tunneling barrier layer (pocket) is added between the source/channel junction to further increase tunneling efficiency of the device at the tunnel junction [35]. Finally, the gate dielectric near the drain region consists of SiO_2_ with low K values, which weakens the gate control of the channel-drain tunneling junction and can effectively reduce the ambipolar current (I_amb_), while HfO_2_ with a high K value near the source region is chosen to increase I_on_ through the method of gate control [36,37].

The structures of this paper are as follows: Section 2 incorporates the proposed device structure along with the design parameters, simulation models, and fabrication process. Section 3 is dedicated to the mechanism, characteristics, and analog/RF performance of the proposed devices (HJ-HD-P-DGTFET). The device parameter variability analysis of the proposed structure is also covered in this section. Furthermore, a performance comparison between Si DG-TFET and HJ-HD-P-DGTFET is also carried out in this section. Finally, Section 4 summarizes the conclusions of this work.

## 2. Device Architecture, Parameters, and Simulation Methods

A schematic diagram of the HJ-HD-P-DGTFET structure is shown in Figure 1, which has two symmetrical gate regions to increase the I_on_. In the proposed TFET, Si_0.6_Ge_0.4_(source)/Si(channel) heterojunction is used to increase the drain current. The source/drain regions are symmetrically located on both sides of the channel, and the p-Si channel is inserted below the gate. Doping is performed in such a way so as to obtain maximum I_on_ current. Here, we doped the p^+^ source region with 1 × 10^20^ atoms/cm^3^, n^+^ drain region with 1 × 10^18^ atoms/cm^3^, n^+^ pocket region with 1 × 10^18^ atoms/cm^3^, and the intrinsic channel with 1 × 10^16^ atoms/cm^3^. Moreover, a conventional Si-DGTFET is used for comparison. All of the device parameters used in the simulation process are given in Table 1.

The device simulations are carried out using a Synopsys Sentaurus device simulator, which solves Poisson’s equation self-consistently with the carrier current continuity equations. The nonlocal BTBT model is used in the simulations to take spatial variation in the energy band into account, and it also considers that the generation/recombination of the opposite carrier type is not spatially coincident. Therefore, the BTBT tunneling process is modeled more accurately. The Shockley–Read–Hall (SRH) and Auger recombination models are considered to include the effect of carrier recombination. The band gap narrowing model and Fermi–Dirac statistics are included because the source regions are highly doped. Moreover, the concentration-dependent and field-dependent mobility models are also adopted in the simulations.

For validation of the models used in the Sentaurus TCAD tool, we calibrated our model against the work carried out by Narang R et al. [38]. Figure 2 shows our model calibration against the work carried out by Narang R et al. [38]. Good agreement between the simulated data and the reported results was observed. The inset shows the simulation result of the transmission characteristics of the HJ-HD-P-DGTFET. It is observed that the I_on_ of the HJ-HD-P-DGTFET can reach 7.79 × 10^−5^ A/μm, and the I_off_ is only 8.16 × 10^−18^ A/μm. Additionally, the subthreshold swing (SS) of the device is only 19 mV/dec.

The proposed HJ-HD-P-DGTFET structure can be fabricated using similar steps as reported for DGTFET [1,2]. The possible fabrication steps of the proposed HJ-HD-P-DGTFET are shown in Figure 3. The process begins by preparing the silicon substrate; then, the n^+^ drain doping region is introduced by vertical As implantation and annealing. Afterward, a thin layer of the n^+^ pocket is grown via epitaxy. Next, the source region is implanted, as shown in Figure 3a. Following this, the channel region is formed by etching. Subsequently, the isolation oxide is deposited to prevent the drain region from etching, as shown in Figure 3b. In Figure 3c, SiO_2_ can be grown on the channel using an oxidation process. In Figure 3d, the SiO_2_ gate dielectric is selectively etched away at the source side by using buffered hydrogen fluoride (BHF) solution. The SiO_2_ gate dielectric at the drain side is protected by photoresist masks. Part of the SiO_2_ gate dielectric layer will be replaced by high-k material in Figure 3e. For the next step, atomic layer deposition (ALD) of high-k material HfO_2_ is performed to fill the gap. After this, HfO_2_ is deposited by ALD on the silicon surface to form a heterogeneous gate dielectric. In Figure 3f, the remaining processes are similar to conventional vertical TFET, involving gate deposition, silicon exposure, etc. [39,40].

## 3. Results and Discussion

This section analyzes the comparison between Si-DGTFET and HJ-HD-P-DGTFET. Additionally, the effects of device parameters on the transfer characteristics are analyzed, such as the Si_1-x_Ge_x_/Si heterojunction, heterogeneous gate dielectric, and pocket. Finally, the RF performance of HJ-HD-P-DGTFET is also analyzed.

### 3.1. The Physical Characteristics of Si-DGTFET and HJ-HD-P-DGTFET

Figure 4a compares the transfer characteristics of HJ-HD-P-DGTFET and conventional Si-DGTFET. Due to the use of a SiGe/Si heterostructure and auxiliary tunneling barrier layer (pocket) of n^+^-doped material, which can increase the tunneling rate, the HJ-HD-P-DGTFET reaches an on-state current (I_on_) of 8.46 × 10^−5^ A/μm compared with 2.83 × 10^−6^ A/μm for the conventional Si-DGTFET. It is observed that the I_on_ for HJ-HD-P-DGTFET is increased by nearly two orders of magnitude compared to conventional Si-DGTFET. Furthermore, a minimum subthreshold swing (SS_min_) of 19 mV/dec and an average subthreshold swing (SS_avg_) of 28.4 mV/dec are obtained, while the SS_avg_ of the conventional Si-DGTFET is 49.9 mV/dec. As a result, HJ-HD-P-DGTFET has obvious improvements in I_on_ and subthreshold swing compared to conventional Si-DGTFET. Finally, the ambipolar effect for HJ-HD-P-DGTFET is noticeably suppressed, as shown in Figure 4a. Figure 4b shows the on-state energy band condition of both HJ-HD-P-DGTFET and Si-DGTFET. It can be observed that in the on-state, the tunnel barrier of HJ-HD-P-DGTFET is evidently narrower than conventional Si-DGTFET at the channel and source junction.

### 3.2. Si_1-x_Ge_x_/Si Heterojunction

As discussed in Section 2, the source region formed by Si_1-x_Ge_x_ is used to improve the on-state current in the HJ-HD-P-DGTFET. It is essential to study the variation in the current value with the different germanium compositions in Si_1-x_Ge_x_, where Si_1-x_Ge_x_ (x = 1) equals Ge. Figure 5a depicts the effect of different Ge compositions in the Si_1-x_Ge_x_ source region on transfer characteristics, and it is very clear that the on-state current of the proposed device increases with the increase in x until x = 1. When the mole fraction of germanium is 0.1, the on-state current is approximately 1 × 10^−5^ A/μm. Furthermore, the on-state current reaches about 8 × 10^−4^ A/μm when the mole fraction is increased to 0.9. However, the increase in the I_off_ is much greater than the increase in the I_on_ when the mole fraction is greater than 0.4. Figure 5b shows the effect of the composition ratio x of Si_1-x_Ge_x_ on the on-state energy band diagram of HJ-HD-P-DGTFET. As can be seen from this figure, the energy valley value of the conduction band decreases with an increase in the germanium composition. Tunneling is defined as the injection of electrons from the source valence band to the channel conduction band. The smaller energy valley value of the conduction band in the source region represents the larger band-to-band tunneling rate without varying other conditions. Thus, x = 0.4 in Si_1-x_Ge_x_ makes a shorter tunneling width compared to other compositions. Consequently, x = 0.4 is chosen as the optimal value in Si_1-x_Ge_x_ to ensure a lower off-state leakage current and a higher on-state current.

Figure 6a–d show the transfer characteristics, on-state energy band diagram, electric field, and potential with different doping concentrations in the source region (N_s_) when the mole fraction of germanium in Si_1-x_Ge_x_ is 0.4. It is observed that the off-state current is decreased, while the on-state current increases as a result of higher source region doping concentration N_s_. However, the on-state current slightly decreases when N_s_ increases from 1 × 10^20^ cm^−3^ to 8 × 10^20^ cm^−3^. It is evident that the maximum value of the I_on_ occurs at a concentration of 1 × 10^20^ cm^−3^, and the corresponding I_on_ is 1.05 × 10^−4^ A/μm, as shown in Figure 6b. It is also observed that the tunneling distance at the point of the tunnel junction decreases with an increase in the source region doping concentration N_s_. Furthermore, the valence band increasing significantly in the source region leads to a smaller tunneling distance when N_s_ is 8 × 10^20^ cm^−3^. Meanwhile, there is a very large tunneling distance when N_s_ is 5 × 10^18^ cm^−3^, which diminishes electron BTBT rates. The value of the electric field under the pocket and channel (–10 nm–10 nm) increases as a result of higher source region doping concentration N_s_, as seen in Figure 6C. Consequently, the increasing electric field under the pocket can help to improve the tunneling probability in this region, while increasing the electric field under the channel will raise the barrier height in the drain/channel interface. However, it is observed in Figure 6b that the conduction band spike at the SiGe/Si heterojunction interface appears when the N_s_ is more than 1 × 10^20^ cm^−3^, which leads to the carriers that pass through barriers being recombined with the opposite carriers at the interface. This results in the on-state current slightly decreasing when the source region doping concentration N_s_ exceeds 1 × 10^20^ cm^−3^. Therefore, the maximum current is attained at N_s_ = 1 × 10^20^ cm^−3^.

### 3.3. Gate Heterogeneous Dielectric Structures

Figure 7a shows the transfer characteristic curve of the HJ-HD-P-DGTFET with heterogeneous gate dielectric structure (HfO_2_ + SiO_2_ or SiO_2_ + HfO_2_) and single gate dielectric material (HfO_2_ or SiO_2_). As shown in Figure 7a, it was observed that the I_amb_ of HJ-HD-P-DGTFET with a heterogeneous gate dielectric structure (HfO_2_ + SiO_2_) and the single gate dielectric material with SiO_2_ decrease by about three orders of magnitude compared with heterogeneous gate dielectric structure (SiO_2_ + HfO_2_) and the single gate dielectric material with SiO_2_ when V_gs_ = −1 V. This indicates using SiO_2_ for the gate oxide layer near the leakage region effectively inhibits the ambipolar behavior of the device in the heterogeneous gate dielectric structure (HfO_2_ + SiO_2_). The I_on_ of the HJ-HD-P-DGTFET with heterogeneous gate dielectric structure (HfO_2_ + SiO_2_) increases by four orders of magnitude compared with the single gate dielectric material (SiO_2_) when V_gs_ = 0.5 V, indicating that the gate oxide layer using HfO_2_ near the source region can effectively improve the open current of the device. Therefore, the combined heterogate dielectric structure of HJ-HD-P-DGTFET can not only suppress the ambipolar behavior but also improve the on-state current.

Figure 7b shows the transfer characteristics of HJ-HD-P-DGTFET with different gate dielectric thicknesses (T_o_). As can be seen from the figure, the I_on_ of HJ-HD-P-DGTFET decreases with increasing gate oxide thickness. The maximum I_on_ of HJ-HD-P-DGTFET decreases from 1.38 × 10^−4^ A/μm to 1.18 × 10^−5^ A/μm when the gate dielectric thickness varies from 1 to 6 nm. Furthermore, the ambipolar current of HJ-HD-P-DGTFET with T_o_ = 1 nm increases by two orders of magnitude compared with T_o_ = 6 nm. Figure 7c explains Figure 7b from the electric field distribution with different gate dielectric thicknesses. It is observed that the maximum electric field at the source/pocket interface (−10 nm to −6 nm) increases with decreases in gate dielectric thicknesses, thus reducing the tunneling distance and improving the on-state current. In addition, the electric field at the drain/channel tunnel junction (10 nm) decreases with decreases in gate dielectric thickness, which weakens the band bending around the drain/channel tunnel junction and also reduces the tunneling window. Consequently, the tunneling probability of the carriers is reduced. and the I_amb_ is also weakened with the increases in gate dielectric thickness at the drain/channel tunnel junction. The larger ambipolar current at T_o_ = 1 nm is due to the interface trap at the gate oxide layer, which results in strong electric field bending. Considering the results above and the process conditions, a gate oxide thickness of 2 nm is considered more appropriate. Figure 7d shows the transfer characteristics of HJ-HD-P-DGTFET with different gate work function (Φ_M_). Firstly, the selection of Φ_M_ is critical for the I_off_, which increases rapidly with decreases in Φ_M,_ as depicted in Figure 7d. Secondly, the I_on_ of HJ-HD-P-DGTFET increases slowly with decreases in Φ_M_. Hence, taking into account the two factors, the optimal value of Φ_M_ is chosen as 4.3 eV.

### 3.4. Auxiliary Tunneling Barrier Layer (Pocket) Optimization Results

In order to improve the performance of the device, we introduced a structure with a lightly doped pocket near the source region in the HJ-HD-P-DGTFET. The proposed device is optimized for pocket length (L_P_) by varying its value from 0 nm (without pocket) to 8 nm. When L_P_ was varied in the channel region, the height of the pocket (T_p_) was kept constant at 20 nm. All the other design parameters are the same as listed in Table 1.

Figure 8 shows the effect of the pocket structure parameters on device performance. It can be seen from Figure 8a that the pocket can effectively improve the transfer characteristics of the device. The I_off_ shows a slight dependence on pocket length, while I_on_ and SS values are independent of pocket length L_P,_ as shown in Figure 8a. It is clear that there is no obvious change in I_on_, but I_off_ slightly increases as L_P_ increases from 1 nm to 8 nm. The I_off_ increases from 2.4 × 10^−18^ A/μm to 8.5 × 10^−16^ A/μm with the increase in L_p_ from 1 nm to 8 nm. As seen in Figure 8b, the BTBT tunneling rate at the source channel junction is reduced with L_P_ and extends towards the channel side, which eventually reduces I_off_ with L_P_ in the device. Therefore, 2 nm is considered as the optimum length of the pocket in order to obtain a lower I_off_ and good process conditions.

Figure 8c shows the influence of the height of the pocket (T_p_) on the performance of the HJ-HD-P-DGTFET, under the condition that other parameters remain constant. It is observed that there is no significant change in I_on_ when the T_P_ increases from 4 nm to 16 nm, keeping the L_P_ fixed at 2 nm. However, the I_on_ increases when the T_P_ increases up to 20 nm. This is due to the fact that the higher height of the pocket results in a larger tunneling surface. That is to say, increasing height of the pocket enhances the device characteristics. Thus, 20 nm is regarded as the optimal height of the pocket.

The variations in transfer characteristics with different pocket doping concentrations (N_p_) are shown in Figure 8d. It is seen that higher pocket doping concentrations (N_p_) cause a higher off-state current. This can be attributed to electric field distribution changes with different pocket doping concentrations, as shown in Figure 8e. The electric field in the source/channel interface (−10 nm) increases with increasing N_p_. Consequently, the increasing electric field near the source region helps to improve the tunneling probability in this region. Therefore, 1 × 10^18^ cm^−3^ is regarded as the optimal N_p_.

### 3.5. Comparison of Analog/RF Performance

Figure 9a shows a comparison of C_gd_ and C_gs_ of the four devices at a frequency of 1.0 × 10^6^ Hz. There is not much difference in C_gs_ for the four studied TFETs, as shown in Figure 9a. Moreover, it is very clear that both C_gg_ and C_gd_ of HJ-P-DGTFET Hk (single gate dielectric material is HfO_2_, and other parameters are the same as HJ-HD-P-DGTFET) and HJ-HD-P-DGTFET increase rapidly with increasing V_gs_, while the C_gg_ and C_gd_ of HJ-P-DGTFET Lk (single gate dielectric material is SiO_2_, and other parameters are the same as HJ-HD-P-DGTFET) and Si-DGTFET (parameters are referred to in Table 1) maintain a very small value. Although the gate capacitance characteristics of HJ-HD-P-DGTFET are poor, capacitance is only one aspect that must be considered, and transconductance and I_on_ will also greatly affect device performance. For TFET, increasing the I_on_ benefits from reducing the tunneling barrier width, but this is bound to increase the capacitance. Therefore, it is necessary to comprehensively consider its DC characteristics and capacitance characteristics. In fact, the large leakage capacitance of the TFET is attributed to its intrinsic capacitance, which is also related to the conduction mechanism. It is difficult to reduce the gate leakage intrinsic capacitance of the TFET, owing to the fact that the gate leakage intrinsic capacitance affects the tunneling current of the device. Therefore, solving the TFET gate leakage capacitance is still a challenging task.

In the design of RF applications, the parameters of cut-off frequency (f_T_) and gain-bandwidth product (GWB) are figures of merit (FOMs). Figure 9b,c show the cut-off frequency and gain-bandwidth product calculated by Equations (1) and (2). The results show that the f_T_ first increases with V_gs_ due to an increased g_m_ value. However, it subsequently decreases with V_gs_ due to the increase in C_gd_ and the reduction in g_m_. The decrease in g_m_ can be attributed to the degradation of mobility with the gate field. It is found that the maximum f_T_ of HJ-HD-P-DGTFET is 19.95 GHz, which is greater than the f_T_ of the Si-DGTFET (0.22 GHz). This makes HJ-HD-P-DGTFET more suitable for RF applications.
(1)$$f_{T}=\frac{gm}{2\pi (Cgs+Cgd)}$$
where g_m_ represents transconductance.
(2)$$\mathrm{GBW}=\frac{gm}{2\pi 10Cgd}$$

As shown in Figure 9c, it is observed that a higher GWB of 2.07 GHz is obtained for HJ-HD-P-DGTFET. Furthermore, it is noted that the GWB initially increases with V_gs_ because of the increase in transconductance (g_m_), then decreases as V_gs_ continues to rise, resulting from an increase in C_gd_ and a decrease in g_m_. Comparing the two devices, the GBW of the HJ-HD-P-DGTFET is less than that of the Si-DGTFET.

### 3.6. Comparison of Different TFETs with HJ-HD-P-DGTFET

In order to understand the potential of HJ-HD-P-DGTFET in ultra-low-power applications, Table 2 shows a performance comparison of different TFETs with HJ-HD-P-DGTFET. Compared to other TFETs [41,42,43,44] with different heterogeneous gate dielectric structures, HJ-HD-P-DGTFET has obvious advantages in I_on_, I_off_, and I_on_/I_off_ ratio. This is because of the improved tunneling rate by using p-type doped SiGe material in the source region and auxiliary tunneling barrier layer (pocket) of n^+^-doped material between the source/channel junction. Compared to heterojunction TFETs [32,34,45,46], HJ-HD-P-DGTFET has obvious advantages in I_on_/I_off_ ratio and I_off_. This is due to the reduced I_amb_ and the increased I_on_ by using a low K value near the drain region and a high K value near the source region. Compared to other novel structural TFETs [47,48], HJ-HD-P-DGTFET has obvious advantages in I_off_ and I_on_/I_off_ ratio. By combining the advantages of heterogeneous gate dielectric structures, heterojunction, and auxiliary tunneling barrier layer, HJ-HD-P-DGTFET can provide high operating current and low static power consumption in ultra-low-power applications.

## 4. Conclusions

In this work, a novel HJ-HD-P-DGTFET is proposed, and its electrical characteristics are simulated and analyzed. The structural characteristics, physical mechanisms, performance with different parameters, and analog/RF performance of HJ-HD-P-DGTFET are discussed and studied. Simulation results show that the new design of HJ-HD-P-DGTFET performs well in terms of both switching and analog/RF characteristics compared with the conventional Si-DGTFET. Benefitting from SiGe(source)/Si(channel) heterojunction and the heavily doped pocket within the channel, the I_on_ for HJ-HD-P-DGTFET is increased by nearly two orders of magnitude compared with conventional Si-DGTFET due to the increase in the tunneling rate. Owing to the use of a high-k HfO_2_ gate dielectric in the source region and low-k SiO_2_ gate dielectric in the drain region, which weakens the gate control of the channel-drain tunneling junction, the I_amb_ is clearly suppressed, and the I_on_/I_off_ ratio is greatly improved. Finally, a large I_on_ of 7.79 × 10^−5^ A/μm, a suppressed I_off_ of 8.16 × 10^−18^ A/μm, I_on_/I_off_ of 9.55 × 10^12^, SS_min_ of 19 mV/dec, f_T_ of 19.95 GHz, and GBW of 2.07 GHz can be achieved by HJ-HD-P-DGTFET. Therefore, HJ-HD-P-DGTFET can be a potential candidate for future low-power IC applications.

## Figures and Tables

**Figure 1 micromachines-14-00784-f001:**
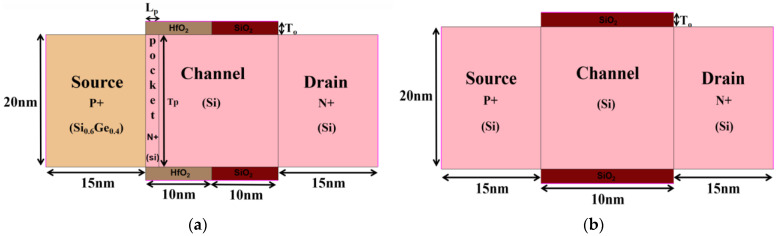
Schematic diagram of (**a**) HJ-HD-P-DGTFET; (**b**) Si-DGTFET.

**Figure 2 micromachines-14-00784-f002:**
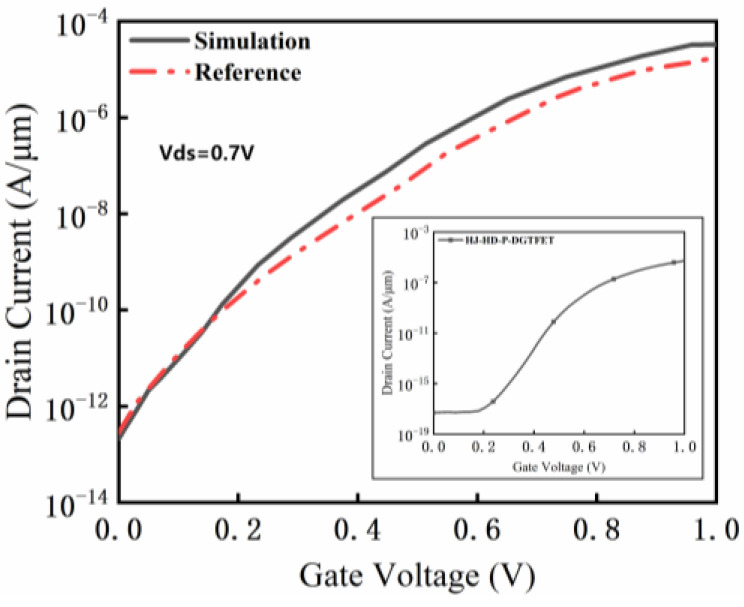
Model calibration against the work from Narang R et al. [38]. The inset shows the transmission characteristics of the HJ-HD-P-DGTFET.

**Figure 3 micromachines-14-00784-f003:**
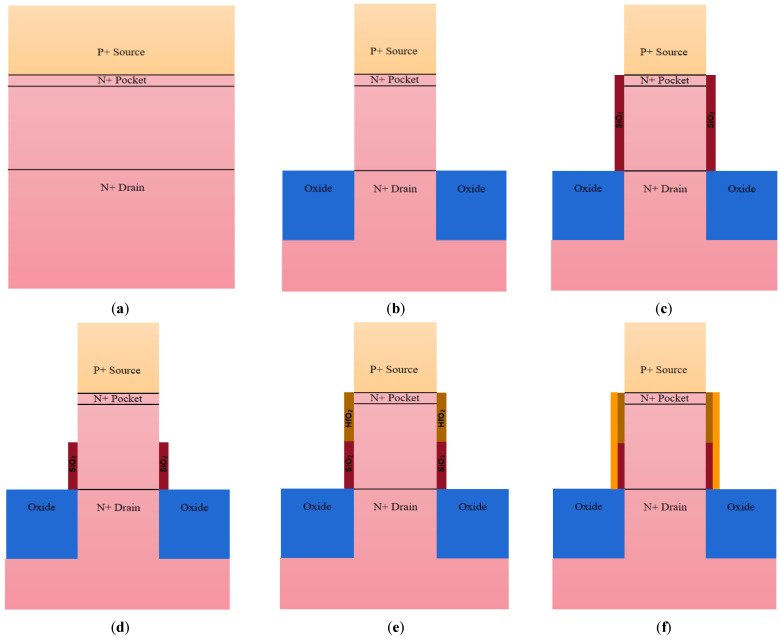
Fabrication process of the HJ-HD-P-DGTFET. (**a**) Silicon substrate preparation and the n^+^ region is introduced, after which a n^+^ pocket is deposited by epitaxy. Subsequently, the source region is implanted; (**b**) etching is performed in the channel region, and isolation oxide is deposited; (**c**) oxidation growth of SiO_2_; (**d**) SiO_2_ gate dielectric is selectively etched away at the source side; (**e**) the HfO_2_ is deposited by ALD; (**f**) gate deposition, silicon exposure, etc.

**Figure 4 micromachines-14-00784-f004:**
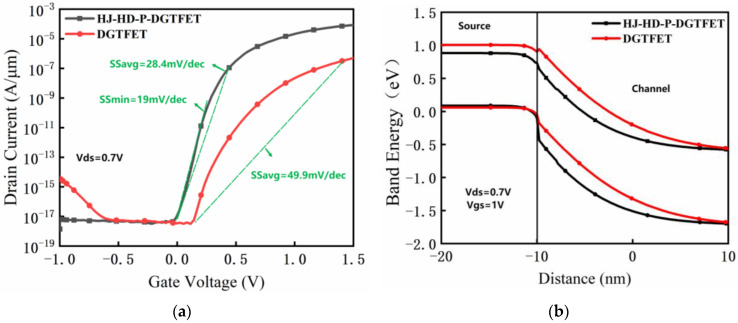
(**a**) Transfer characteristics (**b**) on-state energy band of HJ-HD-P-DGTFET and Si-DGTFET.

**Figure 5 micromachines-14-00784-f005:**
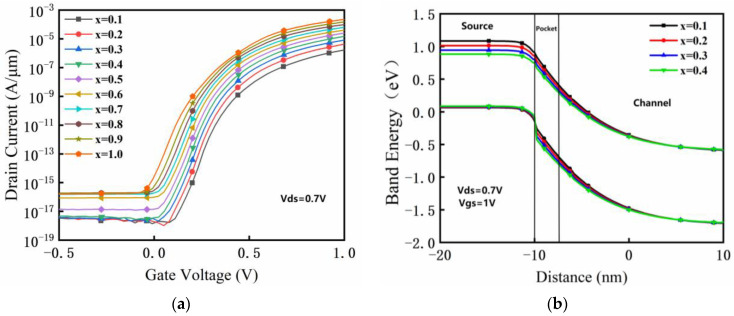
(**a**) The transfer characteristics; (**b**) on-state energy band diagram of HJ-HD-P-DGTFET with different SiGe composition rates.

**Figure 6 micromachines-14-00784-f006:**
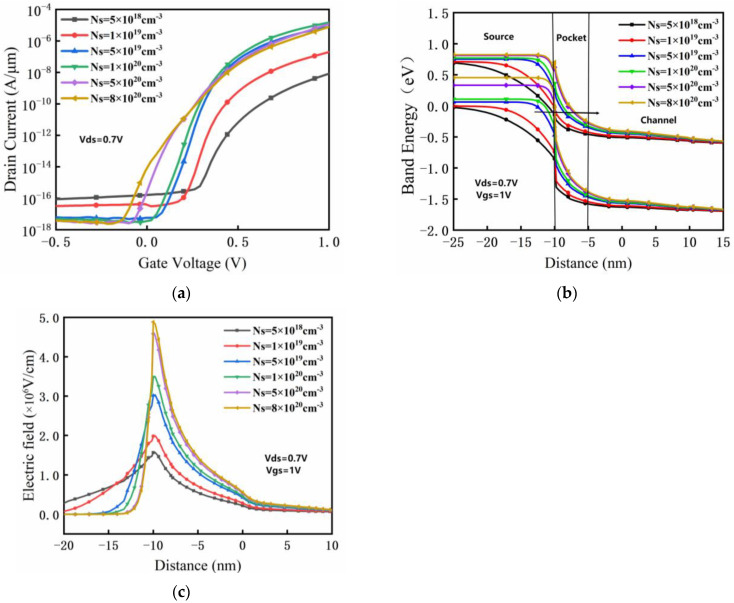
The influence of N_s_ on (**a**) transfer characteristics; (**b**) the on-state energy band diagram; (**c**) the distribution of the electric field.

**Figure 7 micromachines-14-00784-f007:**
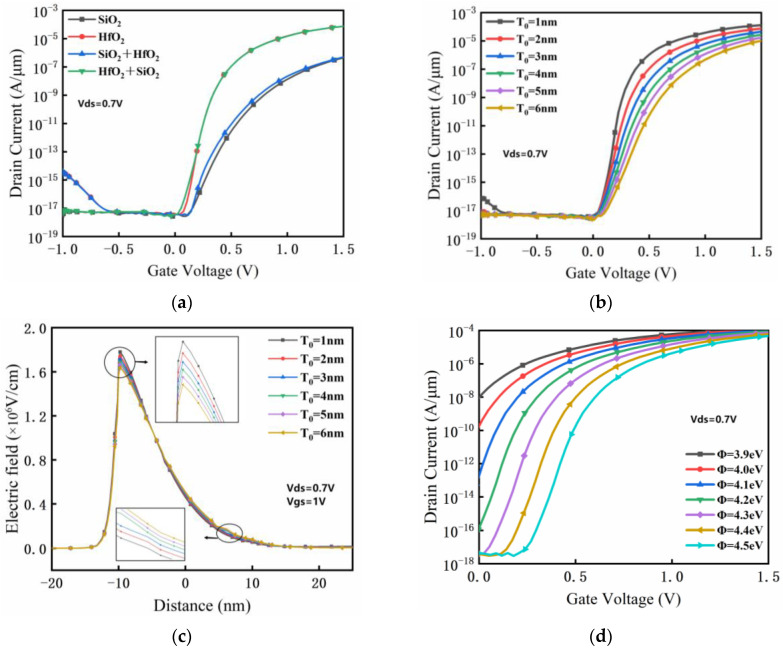
Transfer characteristics of HJ-HD-P-DGTFET with (**a**) different gate dielectric materials; (**b**) different gate dielectric thicknesses T_o_; (**c**) the distribution of electric field with different gate dielectric thicknesses T_o_; (**d**) transfer characteristics of HJ-HD-P-DGTFET with different gate work function Φ_M_.

**Figure 8 micromachines-14-00784-f008:**
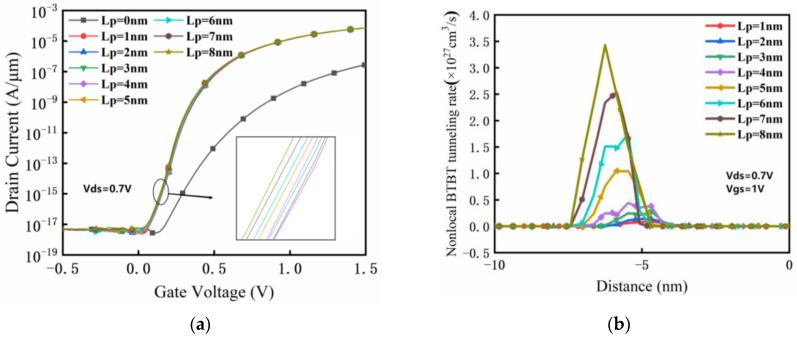
(**a**) Transfer characteristics of HJ-HD-P-DGTFET with different L_p_; (**b**) the distribution of BTBT tunneling rate with different L_p_; (**c**) transfer characteristics of HJ-HD-P-DGTFET with different T_p_; (**d**) transfer characteristics of HJ-HD-P-DGTFET with different N_p_; (**e**) the distribution of electric field with different N_p_.

**Figure 9 micromachines-14-00784-f009:**
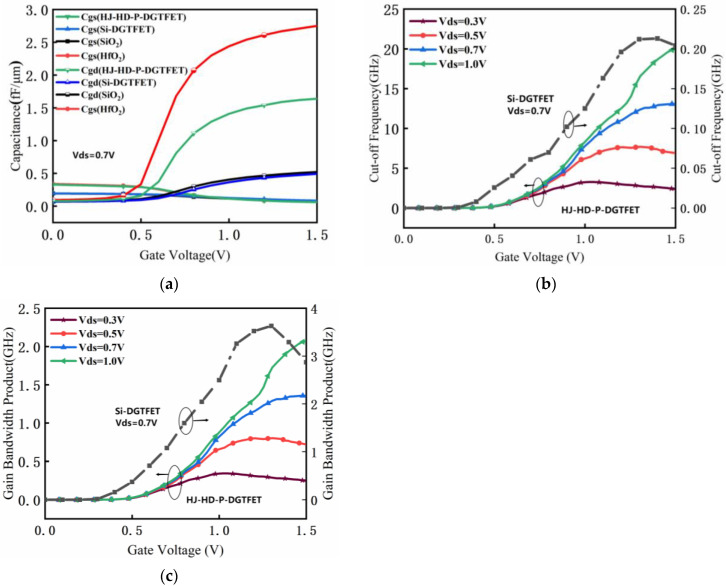
(**a**) Capacitance voltage characteristics of the HJ-HD-P-DGTFET and Si-DGTFET; (**b**) cut-off frequency (f_T_) and (**c**) the gain bandwidth (GBW) plots of both HJ-HD-P-DGTFET and Si-DGTFET.

**Table 1 micromachines-14-00784-t001:** Various structural specification parameters of the presented TFETs for the simulation.

Parameters	HJ-HD-P-DGTFET	Si-DGTFET
Source doping (p-type)	1 × 10^20^ cm^−3^	1 × 10^20^ cm^−3^
Drain doping (n-type)	1 × 10^18^ cm^−3^	1 × 10^18^ cm^−3^
Channel doping (p-type)	1 × 10^16^ cm^−3^	1 × 10^16^ cm^−3^
Pocket doping (n-type)	1 × 10^18^ cm^−3^	–
Gate oxide thickness (T_o_)	2 nm	2 nm
Pocket length (L_p_)	2 nm	–
Gate dielectric constant of SiO_2_ (ε)	3.9	3.9
Gate dielectric constant of HfO_2_ (ε)	25	–
Gate work function (Φ_m_)	4.3 eV	4.3 eV

**Table 2 micromachines-14-00784-t002:** Performance comparison of different TFETs with HJ-HD-P-DGTFET.

Device	I_on_ (A/μm)	I_off_ (A/μm)	I_on_/I_off_ Ratio
CP-DGTFET [41]	5.8 × 10^−7^	10^−15^	10^8^
SP-GDU-DGTFET [42]	10^−7^	10^−17^	10^10^
DF-TFET [43]	3.8 × 10^−6^	2.7 × 10^−17^	10^11^
DS-TFET [44]	1.0 × 10^−8^	1.9 × 10^−16^	10^7^
Mg_2_Si source CP-DGTFET [45]	4.6 × 10^−3^	10^−16^	10^13^
Ge CP-DGTFET [32]	10^−6^	10^−18^	10^12^
Si_0.8_Ge_0.2_ pocket TFET [34]	10^−6^	10^−13^	10^7^
SiGe/Si TFET [46]	6.7 × 10^−6^	2.0 × 10^−10^	10^4^
CS-DL-NT-TFET [47]	1.7 × 10^−5^	2.0 × 10^−17^	10^11^
HD-HJLTFET [48]	4.5 × 10^−5^	1.5 × 10^−16^	10^11^
This work	7.8 × 10^−5^	8.2 × 10^−18^	10^12^

## Data Availability

Data are contained within the article.

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
