# Peer review of "Novel SiGe/Si Heterojunction Double-Gate Tunneling FETs with a Heterogate Dielectric for High Performance"

_micromachines, 2023, doi:10.3390/mi14040784_

Round 1

Reviewer 1 Report

The manuscript by Chen et al. reported a structural engineering of dual gate TFET device using both high-k and low-k for gate dielectric. The topic of this work is interesting and improved device performance has been achieved. However, the feasibility of such heterogate dielectric towards practical fabrication and implementation should be seriously considered. The combination of high-k/low-k dielectric in TFET device has been widely studied (e.g. APL 2010, 96, 122104; Solid-State Electronics 2012, 70, 67; J. Computational Electronics 2017, 16, 30), the demonstrated structure in previous literatures is friendly for practical device manufacturing. But the structure in the present work seems to pose great challenges to fabrication. The authors are strongly encouraged to consider and solve this issue.

Author Response

The manuscript by Chen et al. reported a structural engineering of dual gate TFET device using both high-k and low-k for gate dielectric. The topic of this work is interesting and improved device performance has been achieved. However, the feasibility of such heterogate dielectric towards practical fabrication and implementation should be seriously considered. The combination of high-k/low-k dielectric in TFET device has been widely studied (e.g. APL 2010, 96, 122104; Solid-State Electronics 2012, 70, 67; J. Computational Electronics 2017, 16, 30), the demonstrated structure in previous literatures is friendly for practical device manufacturing. But the structure in the present work seems to pose great challenges to fabrication. The authors are strongly encouraged to consider and solve this issue.

Response 1:  According to the reviewer’ suggestion, the heterogate dielectric towards practical fabrication and implementation has been considered in the revised manuscript. (See:page 3: lines 110-123, page 4: lines 128-133 )

Reviewer 2 Report

The authors have presented work on heterojunction SiGe based double-gate TFET. The paper seems interesting, however, I would recommend the paper given that the authors must clarify a few major concerns. The introduction can include more TFET latest references with SiGe heteromaterial with junctionless or dopingless characteristics such as 10.1109/TED.2020.2971475, 10.1109/TED.2019.2893224, 10.1109/IEDM.2015.7409757. Is the simulated device calibrated with experimental data? If yes, what parameters are tweaked to get the desired output? Is the variation of tunneling mass taken into consideration for proper replication of SiGe? It will be better for the manuscript to focus more on the novelty of the work in the abstract and conclusion. Have the authors' considered defect analysis as the doping conc. is really high which can impede the tunneling process? Why the device shows a kink effect for 8e20 doping in fig. 6(a)? I would recommend correcting the meshing as the electric field curve is not uniform across the source in fig. 6(c). Kindly compare the proposed device with other similar works such as 10.1016/j.sse.2019.107701, 10.1109/TED.2020.2965244, 10.1186/s11671-020-03429-3, 10.1007/s12633-021-01361-4 to show the performance enhancement.

Reviewer 3 Report

In this manuscript, the authors proposed and studied a novel SiGe/Si heterojunction double-gate heterogate dielectric tunneling field-effect transistor with an auxiliary tunneling barrier layer (HJ-HD-P-DGTFET) by using TCAD simulation. Compared to conventional Si-DGTFET, the proposed HJ-HD-P-DGTFET shows higher on-state current and suppressed ambipolar effect indicating the HJ-HD-P-DGTFET is a promising device for low-power consumption radio frequency application. In my view, this manuscript is useful and informative and I would recommend it for publication in Micromachines after minor revision. Since the authors mainly discuss on simulation of their proposed device, it does not make sense to show the fabrication process of this device in Figure 3. Instead, the authors should show schematics of how they set up their simulation in Figure 3.

Author Response

Title: A Novel SiGe/Si Heterojunction Double-gate Tunneling FETs with a Heterogate Dielectric for High Performance

Authors:Qing Chen, Rong Sun, Ruixia Miao, Hanxiao Liu, Lulu Yang, Zengwei Qi, Wei He and Jianwei Li

Journal: micromachines

Manuscript ID: micromachines-2293935

Reviewer #3:In this manuscript, the authors proposed and studied a novel SiGe/Si heterojunction double-gate heterogate dielectric tunneling field-effect transistor with an auxiliary tunneling barrier layer (HJ-HD-P-DGTFET) by using TCAD simulation. Compared to conventional Si-DGTFET, the proposed HJ-HD-P-DGTFET shows higher on-state current and suppressed ambipolar effect indicating the HJ-HD-P-DGTFET is a promising device for low-power consumption radio frequency application. In my view, this manuscript is useful and informative and I would recommend it for publication in Micromachines after minor revision. Since the authors mainly discuss on simulation of their proposed device, it does not make sense to show the fabrication process of this device in Figure 3. Instead, the authors should show schematics of how they set up their simulation in Figure 3.

Response: Thank you for the reviewer’s suggestion. In this manuscript, the device simulations are carried out by using Synopsys Sentaurus device simulator. The main models used in the simulation process are shown in Section 2: Device Architecture, Parameters and Simulation Methods. So we think that it’s not necessary to show schematics of how we set up our simulation. In addition, the purpose of the fabrication process of this device is to increase credibility that the proposed device can be realized in technology.

Round 2

Reviewer 1 Report

The previous concerns have been properly addressed, and it can be published in the present form.

Reviewer 2 Report

I would recommend the work as the authors have clarified most of the concerns.